# Strategy for Fast Decision on Material System Suitability for Continuous Crystallization Inside a Slug Flow Crystallizer

**DOI:** 10.3390/mi13101795

**Published:** 2022-10-21

**Authors:** Anne Cathrine Kufner, Adrian Krummnow, Andreas Danzer, Kerstin Wohlgemuth

**Affiliations:** 1Department of Biochemical and Chemical Engineering, Laboratory of Plant and Process Design, TU Dortmund University, D-44227 Dortmund, Germany; 2Department of Biochemical and Chemical Engineering, Laboratory of Thermodynamics, TU Dortmund University, D-44227 Dortmund, Germany; 3AbbVie Deutschland GmbH & Co. KG, Global Pharmaceutical R&D, Knollstraße, D-67061 Ludwigshafen am Rhein, Germany

**Keywords:** continuous crystallization, microfluidics, slug flow, contact angle, solid–liquid interaction, solubility modeling

## Abstract

There is an increasing focus on two-phase flow in micro- or mini-structured apparatuses for various manufacturing and measurement instrumentation applications, including the field of crystallization as a separation technique. The slug flow pattern offers salient features for producing high-quality products, since narrow residence time distribution of liquid and solid phases, intensified mixing and heat exchange, and an enhanced particle suspension are achieved despite laminar flow conditions. Due to its unique features, the slug flow crystallizer (SFC) represents a promising concept for small-scale continuous crystallization achieving high-quality active pharmaceutical ingredients (API). Therefore, a time-efficient strategy is presented in this study to enable crystallization of a desired solid product in the SFC as quickly as possible and without much experimental effort. This strategy includes pre-selection of the solvent/solvent mixture using heuristics, verifying the slug flow stability in the apparatus by considering the static contact angle and dynamic flow behavior, and modeling the temperature-dependent solubility in the supposed material system using perturbed-chain statistical associating fluid theory (PC-SAFT). This strategy was successfully verified for the amino acids l-alanine and l-arginine and the API paracetamol for binary and ternary systems and, thus, represents a general approach for using different material systems in the SFC.

## 1. Introduction

In recent years, there has been increased research in the area of two-phase flow combined with micro- or minifluidics for a variety of application areas, such as measurement devices in life science and chemistry, as medical devices, microreactors, and heat exchangers, to name a few [1]. A further field of interest is crystallization as isolation technology for the small-scale production of active pharmaceutical ingredients (APIs). A typical production quantity for API production lies in the range of 250–1000 kg a^−1^. Compared to other separation techniques, crystallization processes offer some key benefits, such as adjustable particular product properties and high product purity [2]. Therefore, one or even more crystallization steps are used in more than 90% of API production pathways [3]. The main specifications of the final product are a uniform particle size and shape to ensure constant bioavailability and dosage uniformity [4,5]. Operating modes for crystallization are batch, semi-batch, and continuous. The first mentioned are the most common ones in pharmaceutical crystallization due to the simplicity of apparatuses [4]. However, batch operation has some characteristic drawbacks, such as variability in product quality between batches, encrustation, and high capital costs [5,6]. Therefore, continuous crystallization is advantageous due to high process reproducibility and uniform product quality while operating in a steady state. Furthermore, the higher process efficiency in terms of used substrates, and the use of the same equipment for research and design, as well as for industrial production by the extend of operating time, make the application highly attractive.

A distinction is made between two types of continuous crystallizers for the small-scale production range: mixed-suspension mixed-product removal (MSMPR) crystallizers and tubular plug-flow crystallizers (PFC) [7,8,9]. This work exploits the advantages of a special PFC, the slug flow crystallizer (SFC). The SFC is characterized by a gentle particle treatment with respect to particle suspension and a narrow residence time distribution (RTD) of the liquid and solid phases—basic prerequisites for obtaining a narrow particle size distribution (PSD) and high purity in a reproducible manner during crystallization. In the SFC, managing two immiscible fluids creates a slug flow pattern. Due to easier handling in further downstream and to avoid cross-contamination, gas–liquid segmentation is used in this study. Because of the wall friction, Taylor vortices are induced inside the liquid segments (slugs), which provide increased mixing of the liquid and suspension of particles. Accordingly, crystallization phenomena such as secondary nucleation or agglomeration are reduced.

However, the characteristics and advantages mentioned here only apply if a stable and uniform slug flow is achieved, which depends on many parameters: First, the choice of inner diameter for the tubing is decisive in order to force dominant surface effects and allow the simplified formation of slugs over the entire diameter [10,11]. Therefore, several criteria are postulated for the transition from macro- to mini- and microchannel, but there is no clear definition. Often, the Eötvös number *Eö* (*Eö* < 3.368 [12], *Eö* < 0.88 [13], *Eö* < (2·π)^2^ [14]) or the hydraulic diameter d_h_ [15,16,17,18] are used for confinement into the microscale range.

Second, the consideration of tubing material and material system (solute and solvent) combination is crucial. The property that receives special focus for the configuration of the slug shape is the three-phase contact angle Θ. It provides information on which of the two phases (gas or liquid) the wall-wetting phase is and whether convex or concave slugs are formed. Obtaining convex slugs is desirable in the case of crystallization since the RTD of the liquid phase corresponds to the RTD of the solid phase (particles) present in the liquid. This was demonstrated in our previous publication for crystallizing l-alanine from aqueous solution by using fluorinated ethylene propylene (FEP) as tubing material [19]. For aqueous systems, the utilization of a hydrophobic tubing material, such as FEP, leads to the avoidance of a wall film and the formation of convex slugs in the gas–liquid flow [20,21], whereas hydrophilic tubing material favors the undesirable wall crystallization [20]. In the literature, besides FEP [22,23,24], mostly silicone [20,21,25,26,27,28,29] or polyvinyl chloride (PVC) [20] are used as the tubing material for continuous crystallization applications. A wall film leads to broad RTDs if a material system is combined with a tubing material, where concave slugs are formed (rounded gas bubbles). Since gas bubbles roll over particles at the bottom, particles can be exchanged between neighboring slugs. Particles smaller than or equal to the wall film thickness are especially affected [29]. Accordingly, concave slugs lead to deviations between the RTDs of liquid and particles and negate the advantage of absent axial dispersion. Higher flow velocities increase this effect [29].

Consequently, it is necessary to consider gas–liquid interfacial tension *σ*_G/L_, but also the solid–liquid (*σ*_S/L_) and solid–gas (*σ*_S/G_) interfacial tension. This relationship is described by Θ, which Young [30] defined (Equation (1)).
(1)$$\sigma_{G/L}\cdot \cos\left(\Theta \right)=\sigma_{S/G}-\sigma_{S/L}$$

In the following, Θ serves as a parameter for the suitability of a material system for use in the crystallization process. Based on the literature on two-phase flow and the dependence of Θ on the tubing material and material system used [31,32,33], the wettability is divided into the intervals of highly wetting (Θ < 50°), marginally wetting (50 < Θ < 90°), and poorly wetting (Θ > 90°). A distinction is also made between wet and dry flow patterns [31,34]. The dry flow pattern describes the absence of the wall film and is desirable for crystallization operation, but it also has the disadvantage that the pressure loss is higher due to the higher contact of the liquid with the wall in the apparatus. The dry pattern is not linked to a contact angle range, but its occurrence can be estimated using the capillary number *Ca* (Equation (2)), which implies that for *Ca* < 10^−3^ (under flow boiling situations) [34,35,36] or, respectively, *Ca* < 10^−2^ [37], dry plug flow occurs. *Ca* is determined by the ratio of flow velocity u and dynamic viscosity η with respect to the gas–liquid interfacial tension $\sigma_{G/L}$.
(2)$$Ca=\frac{u\cdot \eta }{\sigma_{G/L}}$$

In general, particularly in further literature dealing with slug flow crystallization, Θ is given based on static contact angle (Θ_stat_) measurements. However, a dynamic equilibrium is established in the apparatus so that the dynamic contact angle (Θ_dyn_) may deviate from Θ_stat_ under certain conditions. For Θ_dyn_, a distinction is made between a receding (at the front of a liquid slug) and an advanced (at the back of the liquid slug) contact angle. If these differ, this is referred to as contact angle hysteresis (CAH). The higher the hysteresis, the higher the probability of an unstable slug flow. This can be forced by surface disturbances or hydrodynamics, for example. Therefore, not only the Θ_stat_ is decisive for obtaining a stable slug flow and for the shapes of slugs, but also the CAH generated in the apparatus itself. Theoretically, by measuring the Θ of the slugs moving in the microchannel, it may be possible to correlate Θ_dyn_ as a function of the fluid velocities, as well as the material properties, and use this model to select the favored combination of the material system and tubing material in the SFC. Due to the difficulties in measuring Θ_dyn_ on such a small scale, as well as the complexity of the multiphase flow, other methods are preferred. Therefore, Θ_stat_ measurements are used in most cases to approximate the dynamic behavior. For instance, in a rectangular tubing for the range of capillary number *Ca* from 10^−6^ to 10^−4^, Skartsis et al. [38] have shown that the dynamic contact angle can be well approximated by the static one. However, this conclusion cannot be directly transferred to all tubing geometries, material systems, and operating conditions. Therefore, Θ_stat_ and Θ_dyn_ should be checked qualitatively for a new material system. 

In combination with crystallization, the choice of a new material system poses a number of additional challenges for operation in the SFC, besides the requirements for slug flow stability, since a high product quality in terms of mean particle size and width of PSD is aimed for, but also a high yield should be maintained. Therefore, besides demands for process safety, which include toxicity, explosion-proof environment, and more, the component’s solubility in the solvent and its temperature dependency are crucial. If the solute solubility in the solvent is very low, the amount of solvent required is very high, and the mass of solids per volume of solvent is minimal. Furthermore, for the application in cooling crystallization, the temperature dependency of solubility is of decisive importance. In addition to other criteria, such as the lowest possible toxicity and chemical stability of the solute in the considered application range, the solvent’s viscosity should be low for good mass and heat transfer.

In conclusion, selecting a new and suitable material system for crystallization in the SFC is challenging and linked to many constraints to maintain slug flow stability and consequently obtain high product quality at the end of the apparatus. Therefore, this work aims to present a systematic approach to decide material system suitability as fast as possible and with low experimental effort in order to enable the continuous operation of the desired product using cooling crystallization inside the SFC. This structured procedure includes the selection of a suitable solvent for the desired solid via a screening of different solvents and tubing materials and the evaluation of the suitability for the SFC reviewing Θ_stat_. Furthermore, the suitability of the selected solvent/tubing material combination is validated by examining the dynamic behavior in the apparatus with regard to flow stability. As the last step, the temperature-dependent solute’s solubility in the solvent is modeled and predicted to evaluate the possible yield for crystallization processes. With the help of this strategy, it is possible to ensure the crystallization of a new material system in the SFC within four steps for binary and ternary systems.

(1)Pre-selection of solvents for the crystallization of the desired solid compound(2)Static contact angle measurements(3)Proof of slug flow stability inside the apparatus(4)Solubility modeling

## 2. Substances Used

Several common solvents were considered to select an appropriate solvent for crystallization inside the SFC by Θ_stat_ measurements. This involves the use of ultrapure, deionized, and bacteria-free filtered water (Milli-Q^®^, *σ*_G/L_(298.15 K) = 72.04 mN m^−1^ [39]) with a total organic carbon content of maximal 3 ppb, purified by a Milli-Q^®^ Advantage A10 apparatus of *Merck KGaA*. As one product component, l-alanine (Ala) purchased by *Evonik Industries AG* with a purity of 99.7% was selected as it has similar particulate properties as high-priced APIs. The saturated aqueous solution was set according to the measurements and regressed data of Wohlgemuth et al. [40] (Equation (3)).
(3)$$c^{*}\left(g_{\mathrm{Ala}}g_{\mathrm{solution}}{}^{-1}\right)=0.11238\cdot exp\left(9.0849\cdot 10^{-3}\cdot \vartheta^{*}\left({}^{\circ}C\right)\right)$$

As a further solid compound, l-arginine (Arg, purity > 99%, *Merck KGaA*) was chosen in order to prove the concept transferability with another amino acid. The following solubility equation was used (Equation (4)) to prepare a saturated aqueous solution and is based on gravimetric measurements conducted in this work.
(4)$$c^{*}\left(g_{\mathrm{Arg}}g_{\mathrm{solution}}{}^{-1}\right)=0.089\cdot exp\left(2.57\cdot 10^{-2}\cdot \vartheta^{*}\left({}^{\circ}C\right)\right)$$

Since the water solubility of APIs is usually low, leading to limited bioavailability [41], paracetamol (APAP, acetaminophen according to USP, > 99%, *Merck KGaA*) was used as a third solid compound to demonstrate the application field of APIs. The regression curve of the saturated aqueous solution was calculated and used according to the measured data from Grant et al. [42] given in Equation (5).
(5)$$c^{*}\left(g_{\mathrm{APAP}}g_{\mathrm{solution}}{}^{-1}\right)=0.0067\cdot exp\left(3.18\cdot 10^{-2}\cdot \vartheta^{*}\left({}^{\circ}C\right)\right)$$

Further solvents studied were ethanol absolute (99.9%, *VWR, σ*_G/L_(298.15 K) = 21.72 mN m^−1^ [39]), 2-propanol (99.9%, *VWR*, *σ*_G/L_(298.15 K) = 21.74 mN m^−1^ [43]), acetone (≥ 99.5%, *Roth*, *σ*_G/L_(298.15 K) = 22.57 mN m^−1^ [44]) and n-hexane (95%, *VWR*, *σ*_G/L_(298.15 K) = 17.78 mN m^−1^ [45]). These (along with water) five solvents are considered first because they are common solvents used for other separation technologies such as extraction processes and are often upstream of crystallization processes [46].

Requirements for tubing material selection are thermal and chemical resistance to a broad spectrum of solvents. For analytical reasons, the transparency of the tubing would be advantageous but not mandatory. Besides FEP, which was successfully used in our previous publications [19,47,48], other materials such as aluminum, glass, polystyrene, and silicone were also tested via the Θ_stat_ measurements for their suitability as a tubing material for the SFC.

Synthetic air (Grade 5.0, *Messer Griesheim*) was utilized as second fluid phase to generate slug flow inside the tubing.

## 3. Modeling of Solubilities Using PC-SAFT Equation of State

According to the presented strategy, the modeling of the solubility within the system under consideration is carried out after selecting the solvent based on Θ_stat_ measurements.

The mole fraction solubility $x_{i}^{L}$ of component i in a solvent (or solvent mixture) was determined by considering an equilibrium between a pure solid phase and a liquid phase according to Prausnitz (Equation (6)) [49]:(6)$$x_{i}^{L}=\frac{1}{\gamma_{i}^{L}}\;\exp\left[\frac{\Delta h_{i}^{\mathrm{SL}}}{RT_{i}^{\mathrm{SL}}}\left(1-\frac{T_{i}^{\mathrm{SL}}}{T}\right)-\frac{1}{RT}\overset{T}{\underset{T_{i}^{\mathrm{SL}}}{\int }}\Delta c_{p,i}^{\mathrm{SL}}\left(T\right)\;dT+\frac{1}{R}\overset{T}{\underset{T_{i}^{\mathrm{SL}}}{\int }}\frac{\Delta c_{p,i}^{\mathrm{SL}}\left(T\right)}{T}\;dT\right]$$

R is the universal gas constant, T is the system temperature, $T_{i}^{\mathrm{SL}}$ is the melting temperature of component i, and $\Delta h_{i}^{\mathrm{SL}}$ is the melting enthalpy at the melting temperature of component i. For amino acids, the difference between the component’s liquid and solid heat capacity $\Delta c_{p,i}^{\mathrm{SL}}$ was recently assumed to be linear dependent on temperature (Equation (7)) [50]:(7)$$\Delta c_{p,i}^{\mathrm{SL}}\left(T\right)=\Delta a_{c_{p,i}^{\mathrm{SL}}}\;T+\Delta b_{c_{p,i}^{\mathrm{SL}}}$$

The combination of a slope $\Delta a_{c_{p,i}^{\mathrm{SL}}}$ and an intercept $\Delta b_{c_{p,i}^{\mathrm{SL}}}$ was used for Ala and Arg in this work. For active pharmaceutical ingredients, such as APAP, the difference between the component’s liquid and solid heat capacity is often assumed to be insensitive to temperature [51]. Thus, APAP was modeled with the heat capacity difference at the melting temperature in this work. All melting properties were taken from the literature and are listed in Table 1.

The activity coefficient $\gamma_{i}^{L}$ of a component i accounts for deviation from ideal-mixture behavior in the liquid phase and is related to the partial derivatives of the residual Helmholtz energy with respect to the mole fraction. Within the perturbed-chain statistical associating fluid theory (PC-SAFT) equation of state, the residual Helmholtz energy $a^{\mathrm{res}}$ is expressed by the sum of a hard-chain ($a^{\mathrm{hc}}$), dispersion ($a^{\mathrm{disp}}$), and association ($a^{\mathrm{assoc}}$) Helmholtz energy contribution (Equation (8)) [55]:(8)$$a^{\mathrm{res}}=a^{\mathrm{hc}}+a^{\mathrm{disp}}+a^{\mathrm{assoc}}$$

These contributions account for repulsion, van der Waals attractions, and hydrogen bonds. Calculation requires the segment number $m_{i}^{\mathrm{seg}}$, the segment diameter $\sigma_{i}$, the dispersion energy parameter $u_{i}k_{B}^{-1}$, the association energy parameter $\varepsilon^{A_{i}B_{i}}k_{B}^{-1}$, the association volume $\kappa^{A_{i}B_{i}}$, and the number of association sites $N_{i}^{\mathrm{assoc}}$ of every component i. $k_{B}$ is the Boltzmann constant. The pure-component parameters used in this work were available in the literature and are summarized in Table 2.

To calculate the segment diameter and the dispersion energy in mixtures of components i and j the combining rules as suggested by Berthelot [59] and Lorentz [60] were used:(9)$$\sigma_{ij}=\frac{1}{2}\left(\sigma_{i}+\sigma_{j}\right)$$
(10)$$u_{ij}=\sqrt{u_{i}u_{j}}\left(1-k_{ij}\right)$$

The binary interaction parameter $k_{ij}$ was introduced for correction of deviations from the geometric mean of the dispersion energies of the pure components and is usually fitted to experimental data of binary mixtures. A linear temperature dependency was assumed in this work:(11)$$k_{ij}=k_{ij,m}T+k_{ij,b}$$

The slope $k_{ij,m}$ and intercept $k_{ij,b}$ for Ala/water, Arg/water, APAP/water, and water/ethanol were taken from the literature and are given in Table 3. The interaction parameters for Ala/ethanol, APAP/ethanol, and Arg/ethanol were fitted to solubility data from An et al. [61], this work, and Jiménez and Martínez [62], respectively, and are also available in Table 3.

The association energy and association volume in mixtures of components i and j were determined by applying the combining rules of Wolbach and Sandler [65]:(12)$$\varepsilon^{A_{i}B_{j}}=\frac{1}{2}\left(\varepsilon^{A_{i}B_{i}}+\varepsilon^{A_{j}B_{j}}\right)$$
(13)$$\kappa^{A_{i}B_{j}}=\sqrt{\kappa^{A_{i}B_{i}}\kappa^{A_{j}B_{j}}}{\left(\frac{\sqrt{\sigma_{i}\sigma_{j}}}{\frac{1}{2}\left(\sigma_{i}+\sigma_{j}\right)}\right)}^{3}$$

## 4. Strategy for Solvent Selection

In order to limit the solvent selection for a specific solid product, in this case Ala, Arg, and APAP, it is important to ensure stable slug flow inside the crystallizer. Furthermore, since the objective is to obtain a high product quality in terms of purity and maintain a narrow PSD, for the latter it is necessary to prevent a wall film (i.e., realize dry plug flow) and, thus, facilitate convex slugs. Therefore, the static contact angle is used in order to select appropriate solvents for crystallization in this apparatus. Afterwards, the dynamic behavior and slug flow stability are proved and correlated to the capillary number *Ca*. Subsequently, the solubility modeling of the selected material system is carried out, which completes the requirements for successful crystallization in the apparatus.

### 4.1. Static Contact Angle Measurements

The Θ_stat_ between solid wall material, liquid, and gaseous phase is crucial for these applications. Therefore, Θ_stat_ measurements were carried out with a drop shape analyzer (DSA30, *Krüss*) equipped with a DS4210 dosing unit and the software ADVANCED to provide initial estimates of the suitability of solvents for SFC application. The measurements are based on the sessile drop method, where a droplet is placed onto the solid surface to be examined by a cannula (*d*_i_ = 0.75 mm). In each case, ten measurements of Θ at the right and left edge of the droplet are measured for three drops (3–20 µL) of a solvent/mixture, and the average value is calculated.

#### Results of Static Contact Angle Measurements

The results of Θ_stat_ measurements of solvents on different tubing materials are presented in Table 4.

Values < 20° indicate that a measurement of a droplet via the software was not possible, caused by the low Θ_stat_. Therefore, the combination of solvent, solid material, and air was assumed to be completely wetted and unsuitable for the use inside the SFC. Using a non-polar solvent (n-hexane) leads to complete wetting of all tested wall materials; hence, n-hexane is not a suitable solvent for SFC. Comparing the surface tensions of ethanol, acetone, and isopropanol (see Section 2), they are in a similar range, and, therefore, the Θ_stat_ are also in a similar range, despite their different solvent nature of polar protic and polar aprotic. However, since Θ_stat_ of isopropanol, acetone, and ethanol are <50° with all materials tested, they are also classified as highly wetting and, thus, not suitable for SFC crystallization. It is becoming clear that water forms higher Θ_stat_ with all the materials tested than the other solvents due to its high $\sigma_{G/L}$. It also can be seen that the most hydrophobic wall material tested, FEP, forms higher Θ_stat_ compared to the other materials. Therefore, the following focuses on FEP as a wall material.

Figure 1 shows the Θ_stat_ of solvent/FEP combinations, which are listed in Table 4. According to the literature, a poorly wetting system (Θ > 90°) is preferred. Consequently, combining an aqueous system and FEP as tubing material is the most suitable setup for crystallization purposes inside SFC for the tested combinations since Θ_stat_ > 100°. This indicates that convex slugs are formed, and the presence of a wall film is negligible.

Conversely, solvents that form a Θ_stat_ < 90° with FEP as the wall material do not appear suitable for use in the SFC. This correlation based on Θ_stat_ measurements is further verified in the next section by reviewing the transport of slugs of all tested solvents in an FEP tubing in order to evaluate the corresponding dynamic behavior.

### 4.2. Proof of Stable Slug Flow Inside SFC

In the following, in order to connect the results from Θ_stat_ measurements and the solubility modeling with the operation inside the SFC and proof of the suitability of the solvent for crystallization purposes, the setup as schematically shown in Figure 2 was applied. The setup can be divided into slug formation and slug flow zone, including image analysis. A solvent, solvent mixture, or saturated solution, which applicability has to be checked, is given inside a feed vessel. For the case of using a saturated solution, those were prepared by setting up a slightly supersaturated solution according to Equation (3), Equation (4), or Equation (5), followed by stirring for 48 h and filtering. The liquid is pumped via a peristaltic pump (*Ismatec Reglo Digital MS-4/12*, *d*_i_ = 2.29 mm Pharmed) to the slug formation zone. This consists of a T-junction (polypropylene, PP) in which the feed is fed in from one side and synthetic air supplied via pipeline pressure from a gas cylinder from the other, forming alternating gas and liquid segments of equal size, respectively. The gaseous volume flow rate was controlled by a high-resolution needle valve (NV-001-HR, *Bronkhorst*) and a flow meter (El-Flow-Select, *Bronkhorst*). The choice of a T-junction is critical for the slug length distribution and its reproducibility throughout the slug flow zone since a squeezing mechanism is evoked as slug formation mechanism, which has already been demonstrated in our previous publication [47]. The inner diameter of the T-junction was chosen to *d*_i_ = 3 mm similar to the tubing’s inner diameter (*d*_i,tubing_ = 3.18 mm) to minimize the slug length variability [66]. After the slug flow was built, the slugs were transported through the FEP tubing (*L*_tubing_ = 7.5 m, *d*_out,tubing_ = 4.76 mm) in the slug flow zone covered with a polyvinyl chloride (PVC) cooling jacket (*d*_i_ = 15 mm), which is filled with deionized water. As no cooling crystallization experiment was performed in this study, no cooling profile was adjusted, and the experiments were conducted at ambient temperature (ϑ_amb_ ≈ 22 °C).

At the end of SFC tubing, a camera (*Samsung NX 300*, 18–55 mm lens) is placed in order to evaluate the slug shape and slug length distribution by image analysis. Therefore, the process tubing is placed in a glass box (14 cm × 6 cm × 6 cm) filled with degassed water at ambient temperature. The back of the box is darkened with black cardboard and a LED lamp (*LED Panel Light* from *LED Universum*) is placed above the box for indirect illumination. This procedure minimizes reflections and light influences from the environment and creates a high contrast for the evaluation of the resulting videos. The image evaluation is conducted by an in-house MATLAB script, which has already been described in a previous publication [47]. The slug length and CAH are evaluated quantitatively and the slug shape qualitatively.

#### Results of Proof of Slug Flow Stability

Images of slugs during operation with different solvents at the end of the apparatus consisting of an FEP tubing are shown in Figure 3.

In all cases, the gas bubble extends over the entire diameter of the tubing, and stable slug flow is formed independently of the solvent used and its Θ_stat_ described in Table 4. Via image analysis, it was quantitatively demonstrated that for each solvent, slugs of equal size are formed, but differences in slug length occur depending on the solvent applied (Table 5). During experiments, it was observed that this is due to the changing slug mechanism in the slug formation zone. While in the case of n-hexane, isopropanol, acetone, and ethanol, the dripping mechanism was observed to form the slugs, in the case of water, the squeezing mechanism produces the slugs.

The squeezing mechanism is desirable for the reproducibility of the slug length according to the literature [47,67,68] and the desired and set operating conditions of short slugs. In general, the slug length is relevant because it influences pressure drop, mixing, suspension of particles, and heat transfer [47,66,69]. The slug length is influenced by mixer geometry and dimension, mixer material/three-phase contact angle, phase ratio/liquid hold-up, velocity of the phases, properties of the phases, supplying of the phases, and the bubble detachment mechanism [47,70,71]. All these influences have to be considered when designing a new apparatus for a new material system.

However, not only slug length, but also slug shape is crucial for crystallization purposes. According to Figure 3, a significant difference in slug shape can be seen for the different solvents tested. While the gas bubbles are rounded in the case of n-hexane, ethanol, and isopropanol, clear edges can be seen in the case of water. For acetone, a transitional form is visible, showing less curvature of the bubble cap compared to the bubbles for n-hexane, ethanol, and isopropanol slugs. The rounded shape of the gas bubbles can lead to the fact that in the presence of crystals in the liquid phase, the crystals can migrate between neighboring slugs and, thus, significantly broaden the RTD of the solid phase. Based on the curvature of the gas bubbles, it can be seen that for the n-hexane, ethanol, and isopropanol slugs, a wall film is present, whereas the contact of the gas bubble and the wall is increased in the case of acetone and water. Therefore, the slug flow for acetone and water is visually assigned to the dry pattern, which is preferred for crystallization. This assignment is confirmed by considering the *Ca* number for the used solvents (Figure 4). The values for hexane, water, and acetone are higher than the values for ethanol and isopropanol ($Ca_{\mathrm{EtOH}}$ = 2.3∙10^−3^; $Ca_{\mathrm{IPA}}$ = 4.8∙10^−3^; $Ca_{\mathrm{Ac}}$ = 5.9∙10^−4^$;\;Ca_{\mathrm{wat}}$ = 5.8∙10^−4^; $Ca_{\mathrm{hexane}}$ = 7.3∙10^−4^). Since all calculated *Ca* numbers are <10^−2^, the higher limit postulated in the literature [37] is not sufficient to predict the dry pattern in our case. The limit of *Ca* < 10^−3^ [34] is applicable to our purposes in the meantime as hexane has already been excluded for suitability due to its non-polar properties.

Furthermore, it is recognizable that no constant Θ_dyn_ is formed in the apparatus itself, as in the case of Θ_stat_, but different dynamic contact angles Θ_dyn_ are present depending on whether at the bottom or top of the slug, left or right. The sampled Θ_dyn_ from the experiment shown in Figure 3 indicates that the receding contact angle is smaller than the advanced contact angle for rounded bubbles, confirming the literature. In addition, the literature states that as *Ca* increases, the CAH range becomes broader [34]. The measured ranges of receding and advanced contact angles for the cases considered are shown in Table 6.

Although the difference in receding and advanced Θ_dyn_ observed in the experiments differ significantly, no instability of the slug flow has been observed, so the critical CAH has not been exceeded. If the absolute values of the dynamic contact angles are considered, the static contact angles are near the range of the dynamic contact angles for water, ethanol, isopropanol, and hexane slugs. However, this does not apply to the acetone slugs. In this case, the static contact angle is lower than the dynamic ones. Due to the lower curvature of the gas bubble and correspondingly higher Θ_dyn_, it is to be expected that the particles in the slugs can be better transported along. This shows that, in addition to the wetting properties, the hydrodynamics are also decisive and must be taken into account. Furthermore, solvents’ volatility should be checked to avoid the expansion of the gas bubble and the change in slug form during the operation. Summarizing, the combination of estimating the *Ca* number and using the Θ_stat_ appears to be sufficient to have a first indication of the slug shape. A dry pattern is expected for *Ca* < 10^−3^ and Θ_stat_ ≥ 90°.

### 4.3. Proving the Operability of Slug Flow Crystallizer with the Solutes

As described in the introduction, some material systems have already been used for SFC crystallization, mainly amino acids or proteins. A previous publication has already demonstrated the evidence of convex slugs for a saturated Ala/water solution in the crystallization process [19]. The measured Θ_stat_ = 98.45° ± 1.69° and calculated $Ca_{\mathrm{AlaSolution}}$ = 6.3∙10^−4^ for the same operating conditions fulfills the above-mentioned criteria.

To verify the hypothesis to other solid compounds, another amino acid, Arg, which to our best knowledge has not yet been used in the literature for SFC, and APAP are chosen. The solvent selection has already been made in Section 4.1, so for the solutes Arg and APAP water seems ideal as a solvent in an FEP tubing with regard to the slug flow stability and slug shape. The solute influences the contact angle depending on the hydrophobicity of the amino acid side chain and the structure of the molecule [72,73]. The measured Θ_stat_ of the saturated aqueous solutions of both solutes on the FEP tubing material result in Θ_stat,Arg_ = 93.26 ± 1.22° and Θ_stat,APAP_ = 90.61 ± 1.62°. Therefore, both solutes have higher impact on the Θ_stat_ than Ala but are still >90°. The *Ca* numbers were calculated to $Ca_{\mathrm{ArgSolution}}$ = 8.7∙10^−4^ for a saturated Arg/water solution and $Ca_{\mathrm{APAPSolution}}$ = 6.3∙10^−4^ for the saturated APAP/water solution. Consequently, both solute/solvent combinations seem to be suitable for crystallization inside the SFC by fulfilling the criteria so that convex slugs should be formed. Evidence of slug shape and stable slug flow in the apparatus can be seen in Figure 5.

In both cases, stable slug flow was observed over the entire tubing length. The dry pattern, which was already anticipated by the *Ca* numbers, can be confirmed visually for both solutes. Furthermore, the slug shape is convex according to the heuristics, so no wall film is present, and a narrow RTD of liquid and solid phases could be achieved. For evaluation of slug flow stability and slug length reproducibility for the saturated Arg/water solution, the characteristic values of slug length distribution were calculated to *L*_50_ = 10.04 mm and *L*_90-10_ = 0.59 mm. For the saturated aqueous APAP solution, slug lengths of *L*_50_ = 9.82 mm and *L*_90-10_ = 2.24 mm were achieved. Based on the Θ_stat_ measurements and the verification of the dynamic behavior inside the SFC, the combination of the two solutes Arg and APAP, respectively, with water is suitable for crystallization in the apparatus and fulfills the requirements with respect to slug flow stability and slug shape for obtaining a high product quality. For further evaluation of the material systems with respect to the suitability for cooling crystallization, temperature-dependent solute solubilities are modeled for the three presented solutes as a next step.

### 4.4. Solubilities for Binary Systems

Figure 6 shows the modeled temperature-dependent solubilities of Ala, Arg, and APAP in water (a) and ethanol (b) at 0.1 MPa using PC-SAFT compared to experimental data. The solubility lines were modeled with the parameters from Table 1, Table 2 and Table 3. Modeling results and experimental data (from the literature, as well as from gravimetric measurements performed in this work) are in very good agreement. PC-SAFT reveals a high accuracy in predicting solubilities over a wide temperature range.

Above the solubility lines are the two-phase regions in which crystallization occurs if an initially homogeneous mixture is cooled starting below the solubility line across the solubility line. Arg solubilities in water are high, whereas the solubility line strongly depends on temperature. Thus, the Arg/water system is well suitable for cooling crystallization as it enables high yields. This is not the case for the other modeled systems. Ala solubilities in water are high, but the temperature dependence of the solubility line is much less pronounced, so yields are too low. In the APAP/water system, the solubilities are comparatively lower than in the other systems. Moreover, the temperature dependence is again too low to reach sufficient yields when applying cooling crystallization.

To increase the yield in the case of APAP/water for crystallization in the SFC, the application in the ternary system is possible. In order to solubilize, ethanol seems to be useful as an increase in yield due to the higher solubility of APAP in ethanol compared to water. Furthermore, ethanol is suitable as a wash liquid in the downstream processing of Ala [78] and can also serve as an antisolvent in the crystallization process of Ala and Arg in aqueous solutions.

## 5. Consideration of Ternary Systems for Slug Flow Crystallizer Application

In order to extend the possible field of application, the strategy shown above is applied to ethanol/water mixtures. Therefore, solubility modeling of Ala, Arg, and APAP in a ternary system is first performed to determine the influence of different compositions of ethanol/water on solute solubility. Afterwards, it is identified which fractions of ethanol in water are usable for a possible crystallization in the SFC based on Θ_stat_ measurements and *Ca* number calculation.

Figure 7 depicts exemplified the solubilities of Ala from 10 °C to 20 °C, 30 °C, 40 °C, and 50 °C in the ternary phase diagram Ala/water/ethanol at 0.1 MPa. The solubility-reducing effect of ethanol is precisely predicted over the entire temperature range due to the excellent agreement between modeled and experimental solubilities. Modeling was performed based on the parameters from Table 1, Table 2 and Table 3, which were used for the binary systems in Figure 6. No additional parameters were fitted. Thus, PC-SAFT shows a high capability in predicting solubilities in solvent mixtures at different temperatures. This is valid in the same way for the ternary systems of Arg/water/ethanol, and APAP/water/ethanol as shown in Appendix A.

In order to obtain the indication about the slug flow stability in the SFC, Table 7 shows the Θ_stat_ measured for different ethanol/water compositions on FEP, as well as the calculated *Ca* numbers for the respective operating parameters.

From Table 7, it is clear that Θ_stat_ decreases with increasing ethanol fraction. Based on the general definition of non-wetting property for Θ > 90°, the mixtures with *w*_EtOH_ up to 10 wt.-% are suitable to evoke stable slugs for crystallization inside the SFC. The *Ca* number increases with higher velocity and rising ethanol content in the mixture.

Figure 8a shows the calculated *Ca* numbers plotted against the measured Θ_stat_. The green region marks the previous limits from the literature (Θ > 90° and *Ca* < 10^−3^) for a dry pattern. The white regions are the transient regions where data points fulfill only one of the criteria, and, therefore, the slug flow stability at these operating points or for this composition should be experimentally verified. Within the gray region, no stable slug flow should be observable. In our case, only the data points for pure water at volume flow rates of 20 mL min^−1^ and 30 mL min^−1^ are in the green area. None of the mixtures with ethanol content seems to be suitable since they are located in the gray region.

This contradicts the observations from our experiments for stable slug flow. Figure 9 shows exemplarily the slug shape formed in the apparatus at a total volumetric flow rate of *Q*_tot_ = 20 mL min^−1^ for all tested ethanol/water mixtures from *w*_EtOH_ = 0–50 wt.-% (All other pictures are given in the Appendix A). According to our results, there is a broader green region where stable slug flow is possible and dry flow pattern is formed, also with solvent mixtures, as this is the case for all mixtures in Figure 9.

Thus, according to the literature, the suitable region is too small and does not fit to the experimental results. From this, the limits for a stable slug flow for a system and the operating parameters under consideration were modified and given in Figure 8b. The limit of partly wetting region (Θ_stat_ > 50°) can be set as a new limit for the contact angle and *Ca* < 6.3∙10^−3^ is set according to the observations. Thus, the range of application is significantly increased, as the majority of the tested mixtures also lie in the stable slug flow area (green area). Operating points or compositions that do not meet any of the criteria should be avoided for crystallization in the SFC.

However, other experimental observations have emerged that should definitely be taken into account for performing reproducible and reliable crystallization in the apparatus. For higher ethanol content, the front interface appears to be more rounded than for lower ethanol content or no ethanol content (experimental observation), confirming the dependence of the meniscus curvature on the *Ca* number [80].

Furthermore, differences in operation with respect to slug formation could be identified since inside the T-junction a transition state from squeezing to dripping mechanism at the compositions *w*_EtOH_ = 20 wt.-% and *w*_EtOH_ = 30 wt.-%, and at *w*_EtOH_ = 50 wt.-% the undesired dripping mechanism was observed. Consequently, the lower interfacial tension of the mixture with rising ethanol content and, correspondingly, the lower contact angle between wall material (T-junction, polypropylene (PP), Θ_PP/wat/air_ = 102°), liquid, and gas lead to a change in the slug formation mechanism. These observations have also been noticed during the handling of higher total volumetric flow rates in the apparatus. The observed slug formation mechanisms and contact angle hysteresis for different compositions and flow velocities are summarized in Table 7. It can be seen that with increasing flow velocity, the transition from squeezing mechanism to dripping mechanism takes place with decreasing ethanol fraction. Accordingly, the operating range in which the SFC is to operate is decisive for the selection of the maximum permissible ethanol content in the mixture. Consequently, at a volume flow of *Q*_tot_ = 20 mL min^−1^, an ethanol content of up to *w*_EtOH_ = 10 wt.-% is permissible in order to enable continuous crystallization operation with high product quality. At a volume flow rate of *Q*_tot_ = 60 mL min^−1^, on the other hand, an ethanol content should be dispensed with so that a constant slug length and, correspondingly, the same crystallization conditions for each crystal and the condition RTD_L_ = RTD_S_ can be fulfilled. With regard to the static contact angles in Table 7, this means that contact angles from Θ_stat_ = 80° (*w*_EtOH_ = 0.3) are also theoretically usable in the apparatus, but here, the restriction applies whereby the flow velocity should not be excessively high, since in this case, the slug formation mechanism does not meet the specifications.

## 6. Conclusions

A quick decision on the suitability of a new solvent system for obtaining a uniform and reproducible product yield with particles of desired size, a narrow width of particle size distribution, and a high purity inside the slug flow crystallizer is possible with the help of an efficient strategy. In this context, particular attention must be paid to slug flow stability and residence time distribution along the tubing, but also to material system-specific criteria for crystallization. In this study, it was shown for several solid components (two amino acids, one API) that selecting a suitable solvent is simplified by using static contact angle measurements and evaluating a dimensionless parameter. Based on the experimental results in this study, the conventional criteria from the literature for the static contact angle (non-wetting behavior for Θ_stat_ > 90°) or the classification of the flow pattern via the capillary number (*Ca* < 10^−3^) are not sufficient due to the various relationships in the complex two-phase flow (velocity influence, three-phase interactions, slug formation mechanism, solute influence in the system under consideration, and more). However, by combining the static contact angle and the capillary number, the range of a dry pattern can be reliably estimated. For the system under consideration, three areas can be defined: the dry pattern area, which is limited by Θ_stat_ > 50° and *Ca* < 6.3∙10^−3^; the transition range, in which only one criterion is reached, and a check for suitability for slug flow stability should be carried out depending on the compositions or the operating conditions; and the range for which no criterion is reached and crystallization in this composition should be avoided in combination with the operating parameters, since a wet pattern results. By using the ranges defined here for Θ_stat_ and *Ca* number, the possible range of applications for the SFC has been extended.

In addition to these two indicating parameters (Θ_stat_ and *Ca*), the volatility of the solvent and the solute solubility, and its effect on the contact angle, should also be monitored and backed up by viewing the dynamic contact angle behavior inside the apparatus. Further consideration of the solute’s solubility in the solvent via criteria and modeling of the temperature-dependent solubility ensures the prerequisites for successful crystallization in the SFC and reduces the experimental and time effort. However, this is only valid if contamination, and disturbances in the material, for example, can be prevented and uniform operating conditions are maintained. This strategy has been demonstrated for both the binary and ternary systems and, thus, represents a general approach for using different material systems in the apparatus.

## Figures and Tables

**Figure 1 micromachines-13-01795-f001:**
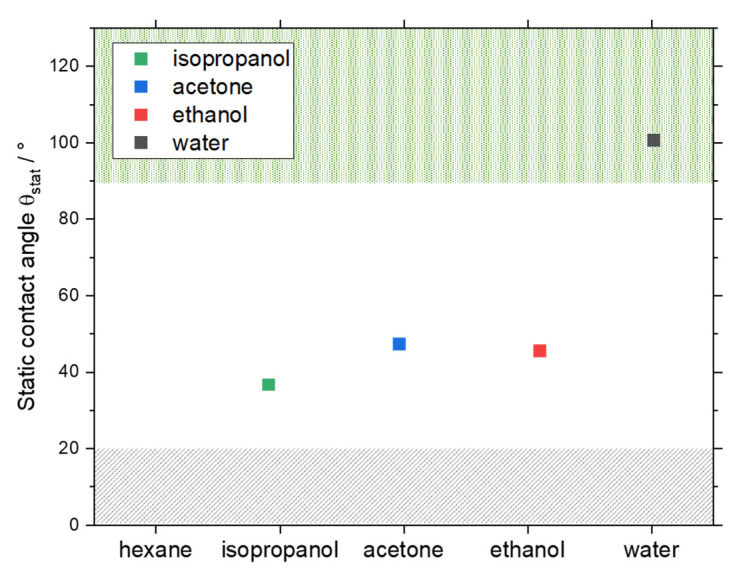
Θ_stat_ for the respective solvents is shown. The grey marked area indicates the region in which Θ_stat_ measurements were not possible (Θ_stat_ < 20°) with the method described before in Section 4.1. The green area (Θ_stat_ ≥ 90°) marks the Θ_stat_ at which a non-wetting behavior is expected, and a stable slug flow might be generated.

**Figure 2 micromachines-13-01795-f002:**
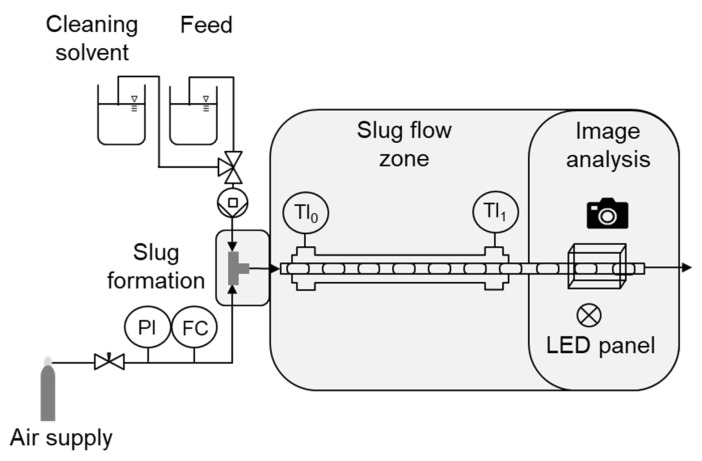
Schematic setup for the validation of solvent suitability for slug flow crystallization.

**Figure 3 micromachines-13-01795-f003:**
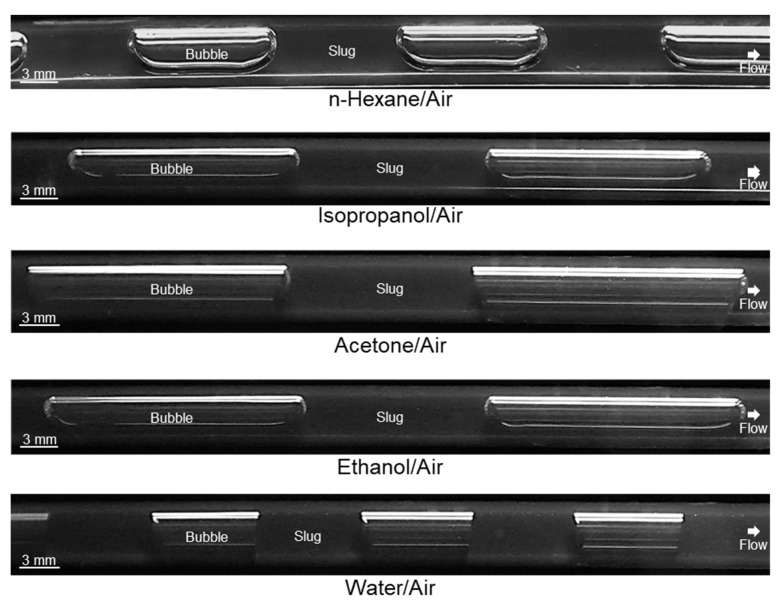
Images of slugs at the end of the apparatus (*L* = 7.5 m) during operation with different solvents in an FEP tubing. The liquid and gas flow rates were set to *Q* = 10 mL min^−1^ each. The experiments were conducted at ambient temperature (ϑ_amb_ ≈ 22 °C).

**Figure 4 micromachines-13-01795-f004:**
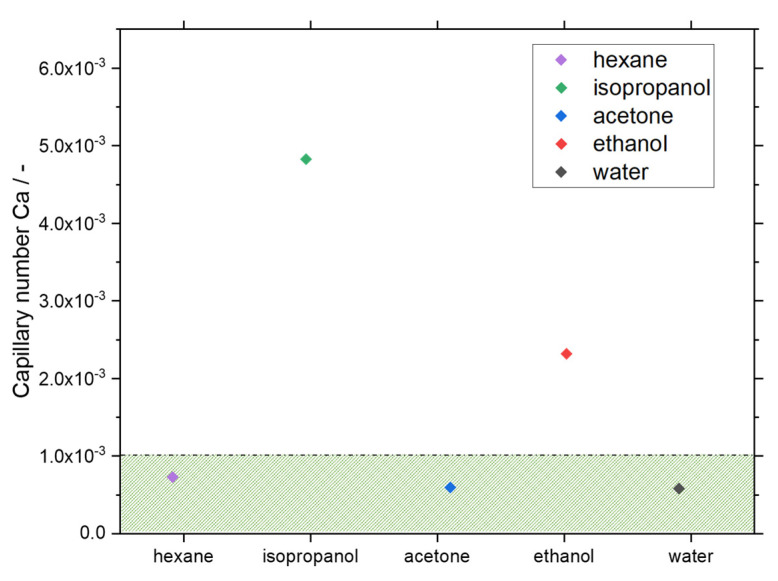
Calculated *Ca* numbers for the tested solvents in the SFC. The calculation was performed for the operating parameters based on the slug flow stability experiments at liquid and gas flow rates of *Q* = 10 mL min^−1^ each and a ambient temperature of ϑ_amb_ ≈ 22 °C. The green area marks the dry pattern slug flow range according to the limit of *Ca* < 10^−3^.

**Figure 5 micromachines-13-01795-f005:**
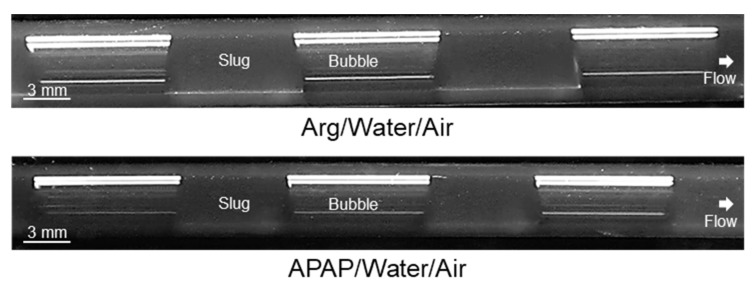
Images of saturated Arg/water (**top**) and APAP/water (**bottom**) slugs at the end of the apparatus (*L*_tubing_ = 7.5 m) during operation inside an FEP tubing of SFC. The liquid and gas flow rates were set to *Q* = 10 mL min^−1^ each. The experiments were conducted at ambient temperature (ϑ_amb_ ≈ 22 °C).

**Figure 6 micromachines-13-01795-f006:**
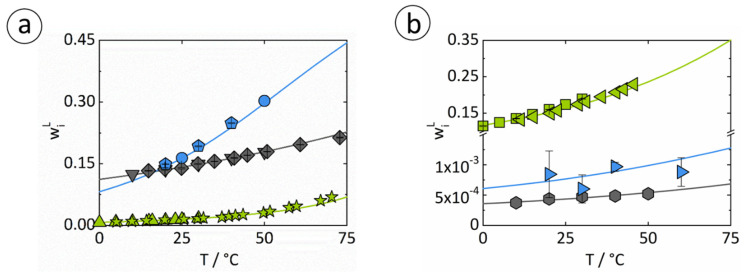
Solubilities of Ala (gray), Arg (blue), and APAP (green) in water (**a**) and ethanol (**b**) at 0.1 MPa: Down-pointing triangles, diamonds, circles, up-pointing triangles, and stars depict measured solubilities in water from An et al. [61], Grosse Daldrup et al. [74], Amend and Helgeseon [75], Granberg et al. [76], and Grant et al. [42]. Hexagons, squares, and left-pointing triangles denote solubility measurements in ethanol from An et al. [61], Granberg et al. [53], and Matsuda et al. [77]. Pentagons and right-pointing triangles are measurements in water and in ethanol performed in this work, respectively. The solid lines are modeled solubility lines using PC-SAFT.

**Figure 7 micromachines-13-01795-f007:**
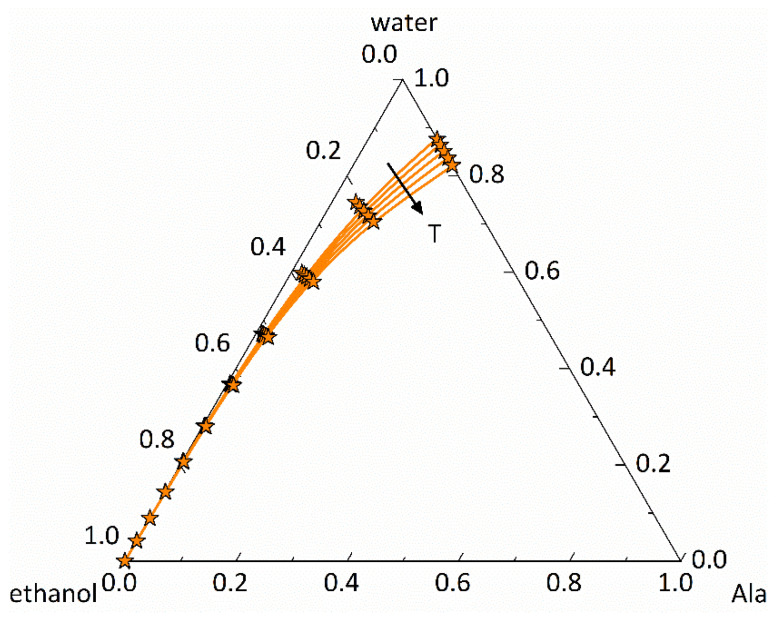
Ternary phase diagram of Ala/water/ethanol at 0.1 MPa with compositions given in mass fractions: Solubility lines were predicted in this work using PC-SAFT, and symbols denote solubility measurements from An et al. [61]. The arrow indicates the direction of increasing temperature from 10 °C to 20 °C, 30 °C, 40 °C, and 50 °C.

**Figure 8 micromachines-13-01795-f008:**
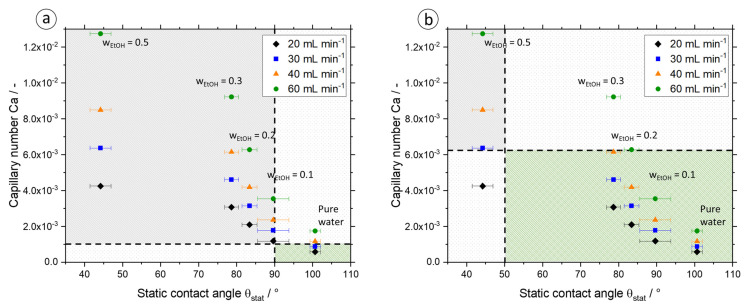
The *Ca* number is plotted against the Θ_stat_ for different EtOH/water compositions and volume flow rates. Delineations for the dry pattern are shown via the black dashed lines based on the literature (**a**) and based on the observations in this work (**b**). The green area marks the dry pattern, the white area the transition, and the gray area the wet region. The latter is unsuitable for crystallization in the SFC.

**Figure 9 micromachines-13-01795-f009:**
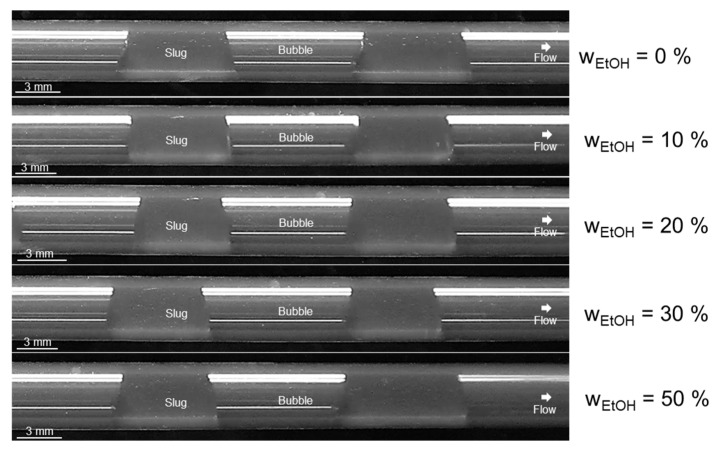
Depiction of the slug flow obtained in the experiments for evaluating the slug shape for different compositions of ethanol/water mixtures at a total volumetric flow rate of *Q*_tot_ = 20 mL min^−1^ at ambient temperature (ϑ_amb_ ≈ 22 °C).

**Table 1 micromachines-13-01795-t001:** Pure-component melting properties of Ala, APAP, and Arg at 0.1 MPa.

Component	$T_{i}^{\mathrm{SL}}/K$	$\Delta h_{i}^{\mathrm{SL}}/\mathrm{kJ}\;{\mathrm{mol}}^{-1}$	$\Delta a_{c_{p,i}^{\mathrm{SL}}}/J\;{\mathrm{mol}}^{-1}\;K^{-2}$	$\Delta b_{c_{p,i}^{\mathrm{SL}}}/J\;{\mathrm{mol}}^{-1}\;K^{-1}$
Ala	608.0 [52]	25.99 [50]	−0.057 [50]	39.923 [50]
Arg	558.0 [50]	32.00 [50]	−0.364 [50]	237.991 [50]
APAP	443.6 [53]	27.10 [53]	0 ^1^	99.800 [54]

^1^ Assumption of $\Delta c_{p,i}^{\mathrm{SL}}\left(T\right)=\Delta c_{p,i}^{\mathrm{SL}}\left(T_{i}^{\mathrm{SL}}\right)$ in this work.

**Table 2 micromachines-13-01795-t002:** PC-SAFT pure-component parameters of Ala, Arg, APAP, water, and ethanol.

Component	$M_{i}/g\;{\mathrm{mol}}^{-1}$	$m_{i}^{\mathrm{seg}}M_{i}^{-1}/\mathrm{mol}\;g^{-1}$	$\sigma_{i}/Å$	$u_{i}k_{B}^{-1}/K$	$\varepsilon^{A_{i}B_{i}}k_{B}^{-1}/K$	$\kappa^{A_{i}B_{i}}$	$N_{i}^{\mathrm{assoc}}$
Ala [56]	89.090	0.0613	2.5222	287.590	3176.600	0.0819	1/1
Arg [56]	174.210	0.0569	2.6572	349.710	2555.450	0.0393	3/1
APAP [57]	151.160	0.0498	3.5080	398.284	1994.200	0.0100	2/2
Water [58]	18.015	0.0669	$\sigma_{\mathrm{water}}$ *	353.950	2425.700	0.0451	1/1
Ethanol [55]	46.069	0.0517	3.1770	198.237	2653.384	0.0320	1/1

* $\sigma_{\mathrm{water}}=2.7927+10.11\;exp\left(-0.01755\;T/K\right)-1.417\;exp\left(-0.01146\;T/K\right)$.

**Table 3 micromachines-13-01795-t003:** PC-SAFT interaction parameters for mixtures of Ala, Arg, APAP, water, and ethanol.

Mixture	$k_{ij,m}/K^{-1}$	$k_{ij,b}$
Ala/water [56]	2.910 × 10^−4^	−0.147962
Ala/ethanol ^1^	1.140 × 10^−3^	−0.3513
Arg/water [56]	0	−0.0145
Arg/ethanol ^2^	2.075 × 10^−4^	−0.134529
APAP/water [63]	1.770 × 10^−4^	−0.051
APAP/ethanol ^3^	1.250 × 10^−4^	−0.08764
water/ethanol [64]	6.860 × 10^−4^	−0.2662

^1^ Fitted to solubility data of An et al. [61] in this work. ^2^ Fitted to solubility data from this work. ^3^ Fitted to solubility data of Jiménez and Martínez [62] in this work.

**Table 4 micromachines-13-01795-t004:** The measured Θ_stat_ for different tubing materials and solvent combinations.

	Θ_stat_/° Glass	Θ_stat_/° Aluminum	Θ_stat_/° Polystyrene	Θ_stat_/° Silicone	Θ_stat_/° FEP
n-Hexane	<20	<20	<20	<20	<20
Isopropanol	<20	<20	<20	<20	36.65 ± 1.37
Acetone	<20	<20	<20	<20	47.33 ± 3.48
Ethanol	<20	<20	<20	23.10 ± 1.34	45.51 ± 3.68
Water	20.89 ± 1.09	66.44 ± 1.41	81.98 ± 0.43	97.99 ± 0.38	100.66 ± 1.37

**Table 5 micromachines-13-01795-t005:** Median slug length *L*_50_ and slug length distribution *L*_90-10_ at the end of the apparatus (*L*_tubing_ = 7.5 m) for the operation with different solvents. The experiments were conducted at ambient temperature (ϑ_amb_ ≈ 22 °C).

	*L*_50_/mm	*L*_90-10_/mm
n-Hexane	10.70	2.23
Isopropanol	24.30	1.27
Acetone	16.37	2.45
Ethanol	20.58	0.85
Water	10.16	0.99

**Table 6 micromachines-13-01795-t006:** Measured receding and advanced dynamic contact angle ranges for the different solvent used in the experiments. CAH_max_ is built by the lowest value for receding and highest advanced dynamic contact angle value for the respective solvent.

Solvent	Θ_dyn_/° Receding	Θ_dyn_/° Advanced	CAH_max_/°
n-Hexane	20–34	25–36	16
Isopropanol	36–63	47–64	28
Acetone	50–60	65–73	23
Ethanol	26–41	40–56	30
Water	82–92	84–92	10

**Table 7 micromachines-13-01795-t007:** Results of Θ_stat_ measurements, observed slug forming mechanism, CAH_max,_ and *Ca* calculation for different ethanol/water mixtures and velocities in the apparatus. The densities, viscosities, and surface tensions for the mixtures were taken from [79] and used for the calculation of *Ca* number at ϑ = 20 °C. The Θ_stat_ measurements were conducted at ambient temperature (ϑ_amb_ ≈ 22 °C) and FEP was used as tubing material.

w_EtOH_/ wt.-%	Θ_stat_/° FEP	Volume Flow Rate/ mL min^−1^	Ca/ -	Flow Pattern Based on Literature Limit (*Ca* < 10^−3^)	Flow Pattern Based on Experiments	Slug Form Mechanism	CAH_max_/ -
0	100.66 ± 1.37	20	5.83∙10^−4^	Dry	Dry	Squeezing	10
		30	8.74∙10^−4^	Dry	Dry	Squeezing	9
		40	1.17∙10^−3^	Wet	Dry	Squeezing	9
		60	1.75∙10^−3^	Wet	Dry	Squeezing	10
10	89.64 ± 4.10	20	1.18∙10^−3^	Wet	Dry	Squeezing	7
		30	1.77∙10^−3^	Wet	Dry	Squeezing	11
		40	2.36∙10^−3^	Wet	Dry	Squeezing	8
		60	3.55∙10^−3^	Wet	Dry	Transition	11
20	83.39 ± 1.96	20	2.09∙10^−3^	Wet	Dry	Transition	12
		30	3.14∙10^−3^	Wet	Dry	Transition	12
		40	4.19∙10^−3^	Wet	Dry	Transition	9
		60	6.28∙10^−3^	Wet	Dry	Dripping	16
30	78.65 ± 1.84	20	3.07∙10^−3^	Wet	Dry	Transition	13
		30	4.61∙10^−3^	Wet	Dry	Transition	10
		40	6.15∙10^−3^	Wet	Dry	Dripping	12
		60	9.22∙10^−3^	Wet	Wet	Dripping	19
50	44.20 ± 2.74	20	4.25∙10^−3^	Wet	Dry	Dripping	13
		30	6.37∙10^−3^	Wet	Wet	Dripping	21
		40	8.49∙10^−3^	Wet	Wet	Dripping	19
		60	1.27∙10^−2^	Wet	Wet	Dripping	17

## Data Availability

All data are contained within the article or the Appendix A.

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
