# Peer review of "Strategy for Fast Decision on Material System Suitability for Continuous Crystallization Inside a Slug Flow Crystallizer"

_micromachines, 2022, doi:10.3390/mi13101795_

Round 1
Reviewer 1 Report
The authors presented a method to enable crystallization of a desired solid product in the SFC as quickly as possible and without much experimental effort. They realized the pre-selection of the solvent/solvent mixture using heuristics, verifying the slug flow stability in the apparatus by considering the static contact angle and dynamic flow behavior, and modeling the temperature-dependent solubility in the supposed material system using PC-SAFT. This investigation is comprehensive and will be of much help in the related applications. I recommend its acceptance.
Minor points: The keyword of "microfluidic" should be "microfluidics"; "PC-SAFT" in the abstract should be defined first.
Reviewer 2 Report
This article developed a successful strategy to decide material system suitability for continuous crystallization inside a slug flow crystallizer. This is a time-efficient strategy which enable crystallization of a desired solid product in the SFC as quickly as possible and without much experimental effort. Meanwhile, the experimental design is reasonable, the experimental data is detailed, and the results are clear, which proves that this method is superior to other existing methods.
Two minor mistakes should be corrected before being publication.
1、 line 283 to 286, in the caption of the figure 1.
I suggest to indicate green area as ?stat > 90° like that described of grey area, which may make a clearer description.
2、 In line 331: the first line should be indented.
